# Shock propagation channels behind the global economic contagion network. The role of economic sectors and the direction of trade

**Zita Iloskics**[1,2,3]*, **Tamás Sebestyén**[1,3,4], **Erik Braun**[1,2,3]

**1** Faculty of Business and Economics, University of Pécs, Pécs, Hungary, **2** ELRN, CERS, Institute for Regional Studies, Pécs, Hungary, **3** EconNet Research Group, UPFBE, Pécs, Hungary, **4** MTA-PTE Innovation and Economic Growth Research Group, Pécs, Hungary

* iloskics.zita@ktk.pte.hu

**Data Availability Statement:** All relevant data are within the manuscript and its Supporting information files.

## Abstract

Examining the spread of macroeconomic phenomena between countries has become increasingly popular after the 2008 economic crisis, but the recent COVID-19 pandemic rendered this issue much more relevant as it shed more light on the risks arising from strongly interconnected economies. This paper intends to extend previous studies in this line by examining the relationship between trade openness and business cycle synchronization. It extends the scope of previous analyses in three areas. First, we use a Granger-causality approach to identify synchronization. Second, trade is broken down to the sector level and third, we distinguish between upstream and downstream connections. These developments allow for a directed approach in the analysis. We use conditional logit regressions to estimate the effect of trade openness on the probability of shock-transmission. The results presented in this study contribute to the literature in two ways. First, in addition to revealing a positive effect of aggregate two-way trade on shock-contagion, it also points out that this overall effect hides diverse behavior in specific trading sectors as well as upstream and downstream channels. Second, while some sectors are not significant channels of shock-transmission in either directions, upstream channels seem to be important in agriculture while downstream channels dominate machinery and other manufactures. Also, there are sectors (chemicals and related products) trade in which affects shock-transmission negatively.

## Introduction

As globalization led to more intensive trade connections between countries, these links became increasingly important in channeling economic shocks between countries. As a consequence, the study of international trade networks gained special attention in the past decades [1–6]. Apart from mapping the topological properties of the global trade network [1, 2], these studies show that structural change within the global trade network is related to the development paths of countries [4, 6] and argue that instability in the global economic system can be attributed to increased globalization and the complexity of the underlying networks [7–9].

Despite the fact that trade connections arise as natural candidates for channels of shock propagation between countries, there is no general consensus in the literature that trade

**Funding:** The research was financed by the Thematic Excellence Program 2020 - Institutional Excellence Subprogramme of the Ministry for Innovation and Technology in Hungary, within the framework of the 4th thematic programme „Enhancing the Role of Domestic Companies in the Reindustrialization of Hungary" of the University of Pécs. Zita Iloskics and Erik Braun has received salary from the above mentioned funding.

**Competing interests:** The authors have declared that no competing interests exist.

relations are either significant [10] or exclusive [11] channels for the spread of macroeconomic shocks. While some studies find that trade relations play an important role in spreading crises [12], others highlight the more important role of financial channels as primary drivers of contagion [13, 14]. Moreover, different channels can affect shock contagion in different ways [15].

The economic consequences of the Covid-19 pandemic shed new light on the role of trade relations in spreading shocks between countries. Due to the sectoral and transportation lockdowns, global supply chains broke, leading to difficulties in sourcing inputs for production from abroad. As a result, trade interconnectedness declined and the structure of bilateral trade linkages changed [16]. Several studies have recently analyzed and modeled the effects of the lockdowns on economic performance [17–21]. [18] show that the extent of economic decline depends on the structure of sectoral linkages, while [17] emphasize that the duration of lockdowns and the number of countries that implement them significantly influence the extent of economic downturn. These studies suggest that bilateral trade links indeed convey economic shocks between countries. Analyzing shock transmission through global supply chains [20] find that job creation in China was significantly affected by demand from those export partners, where lockdowns were in effect.

In the early stages of the Covid-19 pandemic, lockdowns caused a problem primarily in sourcing inputs. As a result, their production has stopped, transmitting shortages to the demand side which again propagate through trade connections [22]. These shock propagation mechanisms are distinguished and interpreted as forward (downstream) and backward (upstream) linkages [23]. The two channels provide significantly different mechanisms to explain the spread of shocks between countries, but the Covid-19 pandemic also points out that both channels may "operate" at the same time.

Early papers investigating the structure of international trade networks and global value chains relied on aggregated bilateral trade data [1, 2, 24–28]. More recent studies, though, made use of sector-level data which allows for a more detailed approach to the analysis of international trade as a complex network as well as to revealing the patterns of shock contagion. These studies contribute to understanding the position of sectors and countries within different production stages [29], the identification of central sectors and sector-level communities [30], the measurement of the nestedness of sectors and countries' within global value chains [31] and the evaluation of sectors' fragility and their role in shock contagion processes [17, 32, 33]. These studies show a meaningful heterogeneity in the role that different sectors play in international trade, which supports the use of sector-level data in mapping the structure of international trade and shock contagion in a more accurate way.

A popular approach to capture shock-propagation between countries is analyzing the synchronization of their business cycles. Also, the analysis of shock-propagation and synchronization from a network perspective gained special interest recently [34–38], with a particular interest in directed connections as in [35, 36, 39]. Several studies have reported increased synchronization of the GDP time series in the previous decades, especially due to the effects of the 2008 financial crisis [34, 35, 37]. Results in [35] show that synchronization across countries does not occur randomly, observed links develop systematically and reflect deeper economic mechanisms. One possible reason behind increased synchronization is globalization and the parallel increase in trade openness. To verify this, several empirical studies have examined the effect of trade openness on business cycle co-movement. According to early studies, there is a moderate positive significant relationship between trade interdependence and cyclical co-movement of macroeconomic indicators [40–42]. As [43] highlights, growth may spill over to trading partners due to increased imports as a result of demand shocks.

More recent studies have confirmed the relationship between business cycle synchronization and commercial relations, but they argue that it is less pronounced than in previous

analyses [44]. These findings were complemented by a number of additional variables, such as monetary and fiscal policy similarities and the degree of financial specialization, that have at least as much effect on the synchronization of business cycles as trade intensity [45]. Also, the relationship between bilateral trade and the co-movement of business cycles differs between subsets of countries [46]: the link is stronger within OECD countries than within non-OECD countries and between OECD and non-OECD countries. Moreover, pairwise synchronization is significantly increased if both countries are industrialized or belong to the group of developing countries [47].

In this study we re-examine this issue between trade openness and synchronization in the spirit of the previous literature [40–43, 45, 47, 48], but we extend the analysis in two ways. First, we use a directed approach distinguishing between upstream and downstream contagion and second, we provide a sector-level decomposition of the shock propagation (trade) channels. While a majority of the studies establish synchronization on the basis of cross-correlation [43, 45, 46, 48–50], or some other undirected synchronization indicator [51], we employ a causality approach as in [34–36, 39] and examine whether trade linkages contribute to the extent to which one country's cyclical behavior affects that of another country. While cross-correlation may falsely identify co-movement if business cycles are driven by unobserved common factors, using causality tests we can focus on events when the economic situation in one country truly affects that in an other country. In this sense, the causality-based approach is better able to identify the spread or contagion of economic shocks from one country to another while cross-correlation only detects co-movement of cycles.

Using a causality approach, on the other hand, allows for mapping directed networks of shock contagion (or synchronization) between countries, which also means that trade connections can be taken into account in a directed manner. This leads us to the final analytical setup of this paper where the probability of shock contagion between countries is associated with the strength of trade connections between them, where the latter can be differentiated according to its direction (upstream and downstream trade linkages) or sector of activity. While previous studies of [43, 46] also used a sector-level approach in analyzing the relationship between trade and business cycle co-movement, our paper extends the analysis at this disaggregated level and contributes to the literature in two ways. First, using causality tests to map shock contagion allows us to take into account synchronization in a directed manner, and second, this can be matched with directed trade data at a relatively detailed sectoral level. This way, we are able to analyze that which economic sectors contribute to shock contagion across business cycles and whether upstream or downstream transmission is more relevant.

## Materials and methods

### The shock contagion network

In this study, we build on previous work of [35] who capture shock propagation through estimating causal relationships between the business cycles of national economies. The analysis is based on output data from the OECD countries except Turkey, plus Bulgaria and Romania for the period between 1996 and 2019. We construct time windows of 52 quarters within this time frame and estimate Granger causality between the cyclical components of country-level GDP series to obtain a network of shock contagion.

The nodes of this directed binary network are the countries and the links between them are present if the GDP-cycle of a country affects that of another country. Using the rolling time windows, we are able to draw this contagion network in a longitudinal fashion through the sample.

While in time series analysis many studies use GDP growth rates to establish macroeconomic co-movement [50, 52], we follow another approach by filtering the cyclical components of GDP level series as in [37, 48]. More precisely, we extract the cyclical component $\hat{y}_{\tau,i}$ of GDP series for all country $i$ and quarter $\tau$ for a sample of 42 countries, using HP-filtering. Then we run pairwise Granger causality tests for all pairs of filtered GDP series, by estimating the following two regression models for every pair of countries $i$ and $j$, with time lag $L$:

$$\hat{y}_{\tau,i} = \beta_0^1 + \sum_{l=1}^{L}\beta_l^1\hat{y}_{\tau-l,i} + \varepsilon_{\tau,i} \tag{1}$$

$$\hat{y}_{\tau,i} = \beta_0^2 + \sum_{l=1}^{L}\beta_l^2\hat{y}_{\tau-l,i} + \sum_{l=1}^{L}\gamma_l\hat{y}_{\tau-l,j} + \mu_{\tau,i} \tag{2}$$

Once these models are estimated, we calculate the F-statistic for all country-pairs as $F_{ij} = (RSS_1 - RSS_2)/RSS_2(N - L)/L$, where $RSS_1$ and $RSS_2$ are the residual sum of squares of models Eqs (1) and (2) respectively, while $N$ is the sample size. Given these test statistics, we calculate the probability of the F-values ($P(F_{ij})$), and establish a contagion link from country $i$ to $j$, labelled as the directed country-pair $c$, as:

$$a_{c,t} = \begin{cases} 1, & \text{for } P(F_{i,j}) < f_{ij}, i \neq j \\ 0, & \text{otherwise} \end{cases} \tag{3}$$

The time index $t$ above refers to the rolling time window on which the estimation is executed: $\tau \in (t, \ldots, t + 51)$. More details of this process, together with the relevant data and codes can be found in [35]. The $a_{c,t}$ values refer to (binary) edges of shock transmission between countries. These edges are labelled as *shock contagion links* later in the paper, while the network is referred to as the *shock contagion network*. Note, that we have a shock contagion network for every time period $t$, i.e. every time window.

While Granger causality tests are widely used in time series analysis as a method to identify casual relationships between observed variables, it is important to emphasise its limitations. Although Granger causality is suitable for detecting co-movement between time series and also reflects the sequence of events, these identified cross-period co-movements does not always represent real causal relationships. The opposing views on the existence of actual causality and the different philosophical approaches to its identification are reviewed by e.g. [53, 54].

## Trade data

Given this network drawn, the paper examines whether the existence of these shock contagion links can be explained by bilateral trade volumes. Data on the latter is extracted from the UN Comtrade database where the trade volume of goods by sectors is fully accessible between 1996 and 2019. The sample covers the OECD countries, except Turkey, but includes Bulgaria and Romania in addition. To bring the trade data in line with the shock contagion networks established for the rolling time windows, trade data was averaged over the corresponding 13 years for all country pairs so that the two sides of our equations are matched in time. Altogether, we have 42 countries and 42 time periods (time windows) available in the dataset.

In the empirical analysis, we decompose total trade into sectors by using SITC Rev. 3 classification. Based on this approach, we can analyze upstream and downstream shock propagation mechanisms in a more detailed manner, by identifying those sectors which play an important

role in the contagion process. It is important to note, however, that a more detailed sectoral decomposition increases the risk of over-specification in the estimation. This means that at a very detailed sectoral level, we can find significant trade only between a few countries which then does not allow for a general conclusion to be drawn for the whole sample. For this reason, we use the first-level breakdown of SITC Rev. 3 in the following analysis, covering 10 distinct industrial sectors. These sectors are listed in Table 1. The sector of animal and vegetable oils, fats and waxes (sector number 4) is the smallest sector with respect to its share in total trade in our sample (see Fig 4 for reference). Also, we found a lot of missing data in the case of this sector for country-pairs and years which may render the regression estimations biased and unreliable. As a result, we decided to exclude this sector from the sector-level analyses in what follows. All the data that was used in the paper is compiled in S1 and S2 Tables.

## The basic model

As it was shown in the previous section, by setting up rolling time windows our dataset on shock contagion and trade gains a time dimension. The units of observation are directed country-pairs, and we have information on shock-contagion and trade between these country pairs for different (overlapping) time periods. This allows for a panel-econometric framework to estimate the effect of trade connections on shock contagion, similarly to [51]. On the other hand, the dependent variable in the analysis is based on the shock contagion network described previously. This network is binary, providing information only on the existence or lack of link between a given country-pair in a given period. As a result, we need a binary panel model specification, estimating the probability that a particular event occurs. The most commonly applied technique is the fixed effects logit model, an important advantage of which is that it is able to control for unobserved heterogeneity through estimating observation-specific fixed effects [55, 56]. The estimation is carried out in the Stata software using the clogit function, which is written as follows:

$$Pr(a_{c,t} = 1 | \mathbf{x}_{c,t}) = P(\alpha_c + \mathbf{x}'_{c,t} \boldsymbol{\beta}), \tag{4}$$

where $a_{c,t}$ is the binary dependent variable, showing the existence ($a_{c,t} = 1$) or lack ($a_{c,t} = 0$) of shock-transmission between a country-pair $c$ in period $t$, $\mathbf{x}_{c,t}$ is the vector of independent variables for country-pair $c$ in period $t$, $\boldsymbol{\beta}$ is the estimated coefficient vector, $\alpha_c$ is the time-independent fixed effect specific to country-pair $c$ and $P(z)$ express the logistic distribution: $P(z) = \{1 + exp(-z)\}^{-1}$.

To examine how trade between country-pairs affects the spread of shocks, the fixed-effects model compares the instances (time periods) of a given country-pair when shock transmission is observed to those instances when it is not observed (ceteris paribus). Subsequently, the average of such differences across all country-pairs determines the average impact of trade on the

**Table 1. Sectoral decomposition used in the analysis.**

| SITC Code | Name | SITC Code | Name |
|---|---|---|---|
| 0 | Food and live animals | 5 | Chemicals and related products, n.e.s. |
| 1 | Beverages and tobacco | 6 | Manufactured goods classified chiefly by material |
| 2 | Crude materials, inedible, except fuels | 7 | Machinery and transport equipment |
| 3 | Mineral fuels, lubricants and related materials | 8 | Miscellaneous manufactured articles |
| 4 | Animal and vegetable oils, fats and waxes | 9 | Commodities and transactions, n.e.s |

spread of shocks. Therefore, the condition for using the model is that each country-pair is included at least twice and the values of the variables change over time [55, 56], which leads to some challenges in the estimation of fixed effects with a binary dependent variable [57, 58].

As the observation-specific fixed effects are identified on the basis of variability in the dependent variable for the separate observations, the procedure can not be used for those observations where the dependent variable does not change over time—a situation which can arise easily with binary data. Indeed, there are 1722 directed country-pairs in our dataset (not counting self-loops), out of which 635 contains only zeros in the dependent variables (we never observe shock-contagion between them) and 173 contains only ones (shock contagion is observed in all periods). This is 47% of the whole sample which is excluded from the estimation. Recent studies also recognize this issue and propose conditions under which it is appropriate to apply the different types of binary models. Results of [57] support the application of the fixed effects logit (conditional logit) model to our data. They show that the conditional logit model produces unbiased coefficients when the number of observations per group is above 30. The groups are the country-pairs in our case, hence we have 42 observations (time periods) per country-pair. On the other hand, the binary event on the left hand side is not a true rare event, because we observe shock-transmission (i.e. ones in the binary dependent variable) in 30% of the cases. By checking the distribution of the observed shock-transmission events within groups (country-pairs), we can see a relatively even distribution excluding the two extreme values, so we consider the use of the conditional logit model to be justified (the histogram of this distribution is found in S1 Fig). Recent studies also call attention on the limitations of applying logistic regression in the case of high-dimensional data [59]. However, this limitation is not binding in this study, since the regression analysis employs 37 variables for approximately 38,000 observations, meaning that the condition for high-dimensionality does not apply. Moreover, these variables are not used simultaneously in the estimations (the most extensive estimation includes 24 variables and runs with 34,152 observations).

## Model variants: Upstream and downstream channels, sectors

We employ conditional logit regression to estimate the relationship between trade connections and shock contagion, under various model setups. These model variants differ in the key variable of interest, but also in the way how different control variables are entered on the right-hand side of the regression equations (the composition of vector $\mathbf{x}_{c,t}$ in Eq 4). First, Table 2 summarizes the different ways that we use to construct trade data: on an aggregate basis, at sectoral decomposition and/or distinguishing between upstream and downstream connections. Later, Table 3 will provide an overview about how these different indicators of trade show up on the right-hand side of the estimated regressions in combination with other control variables.

In the baseline model (Model 1), the key independent variable is bilateral trade in goods between countries. In further models we disaggregate this variable along two dimensions. First, we take into account the direction of trade between countries, distinguishing between export (downstream) and import (upstream) connections (Model 2). Then, we use a sector-

**Table 2. Various explanatory variables and models applied to estimate shock contagion.**

| Trade links are... | Aggregated | Sectorally disaggregated |
|---|---|---|
| Non-directed | $T_{c,t}$ | $T^s_{c,t}$ |
| Directed | $U_{c,t}; D_{c,t}$ | $U^s_{c,t}; D^s_{c,t}$ |

**Table 3. Summary of the different model setups used in the analysis.**

| | | How does trade enter on the right-hand side? | | | | | |
| | | Aggregate trade | | | Sector-level trade | | |
| | | Two-way average ($T_{c,t}$) | Upstream ($U_{c,t}$) | Downstream ($D_{c,t}$) | Two-way average ($T_{c,t}$) | Upstream ($U_{c,t}$) | Downstream ($D_{c,t}$) |
|---|---|---|---|---|---|---|---|
| In what combination do variables enter on the right-hand side? | Only the key variable plus controls | Model 1 | Model 2B-u | Model 2B-d | Model 3B | Model 4B-u | Model 4B-d |
| | Key variable plus opposite direction plus controls | - | Model 2A | | - | Model 5B | |
| | Key variable plus all other sectors in the same direction plus controls | - | - | - | Model 3A | Model 4A-u | Model 4A-d |
| | All sectors in all directions plus controls | - | - | - | - | Model 5A | |

level disaggregation (Model 3) and finally we differentiate both along sectors and trading direction (Model 4 and Model 5).

We use a modified version of the standard trade openness indicator (ratio of imports and exports to GDP) that is applied in most studies (i.e. [43, 49]). Since the dependent variable (shock contagion) is directed by definition (see Eq 3), we also render these openness indicators to be directed in two ways. First, focusing on either exports or imports allows for a directed account for trade in a natural way, and second, by normalizing with only the target country of the shock contagion also adds direction to the openness indicator. This approach reflects that these indicators are going to be regressed on directed shock-contagion and the relative importance of trade connections in the destination country is more important in this respect as that of the origin country. Thus relative trade openness ($T_{c,t}$) of goods between countries was calculated as follows:

$$T_{c,t} = \frac{X_{ij,t} + X_{ji,t}}{2Y_{j,t}}, \tag{5}$$

where $c = i \rightarrow j$ labels a directed country pair, $X_{ij,t}$ is the average trade volume moving from country $i$ to country $j$ over time period (time window) $t$ and $Y_{j,t}$ is the average GDP of country $j$ over time period $t$. Trade flow $X_{ij,t}$ can be interpreted as the export of country $i$ to $j$ or the import of country $j$ from $i$. While in principle these quantities are identical, reported statistics in our data source differ: we used the average of reported exports of $i$ to $j$ and reported imports of $j$ from $i$ as the input values to our indicators as in [47]. As a result, the indicator in Eq 5 reflects relative trade volume (openness) between country $i$ and $j$, irrespective of the direction of trade. However, we normalize absolute trade volume by the GDP of the (shock) destination country. This renders the $T_{c,t}$ indicator, referred to as *relative trade openness* later on, to be directed even if the nominator contains bidirectional trade volumes. This choice of normalization ensures that we rule out size effects in the first place and take into account the relative strength of a given trade relationship within the shock receiver country in the second place. The latter is important as we expect more impact from country $i$ on country $j$ if the given trade link is relatively important in the economy of country $j$. Also, the directed relative trade openness indicators are more easily aligned with directed shock contagion observations.

As mentioned before, the aggregated trade volumes in Eq 5 are first disaggregated along their direction. In other terms, we distinguish trade flows from country $j$ to country $i$ as backward (upstream) trade of country $i$ ($U_{c,t}$), from trade flows from country $i$ to country $j$ as

forward (downstream) trade of country $i$ ($D_{c,t}$):

$$U_{c,t} = \frac{X_{ji,t}}{Y_{j,t}},\tag{6}$$

$$D_{c,t} = \frac{X_{ij,t}}{Y_{j,t}},\tag{7}$$

where notations are the same as in Eq 5.

Eqs 6 and 7 describe two different channels, based on trade connections, through which shock contagion can occur [57]. In the case of both indicators above, $c$ refers to a directed country-pair $i \rightarrow j$. This corresponds to the shock contagion link $a_{c,t}$ in Eq 3, reflecting the observation of shock-transmission from country $i$ to country $j$. Keeping this in mind, $U_{c,t}$ in Eq 6 reflects those trade channels which give rise to an upstream mechanism of shock contagion, where it is the import of country $i$ from country $j$ ($X_{ji,t}$) which transmits the shock from country $i$ to $j$. This is typically a demand-driven channel, where a shock to country $i$ decreases demand for imported products sourced from country $j$. In this mechanism, the shock propagates through the backward or upstream linkages of a country. On the contrary, $D_{c,t}$ in Eq 7 reflects trade channels which give rise to the downstream mechanism of shock contagion where the export from country $i$ to country $j$ ($X_{ij,t}$) channels the shock between these countries. In this case a decline in country $i$'s production spreads over to country $j$ through the export of the former, ultimately causing a decline in economic activity in country $j$ as well. Thus, the shock propagates through the forward or downstream linkages of country $i$.

The three indicators for aggregate ($T_{c,t}$), upstream ($U_{c,t}$) and downstream ($D_{c,t}$) trade openness can be calculated at a sectoral level as well, which is going to be the basis for our detailed sector-level estimations. To sum up, we search for an association between the probability of shock contagion on a country pair $c$ (proxied by $a_{c,t}$) and the intensity of trade as measured by the relative trade openness indicators (overall $T_{c,t}$, upstream $U_{c,t}$, downstream $D_{c,t}$ and possibly broken down to sectors).

In addition to these indicators of trade openness, we include two control variables into our model. First, in the spirit of traditional gravity models, we use the size of the two countries in a given country-pair $c$ measured by their population. This choice is motivated by the assumption that it is more likely to observe shock transmission between large countries/economies. Second, we include the relative development level of the two countries as well, measured by GDP per capita. This allows for measuring the effect of the development or productivity gap on shock transmission.

Table 3 provides an overview of the different model specifications that we use to estimate the role of trade connections in shock propagation. The baseline model (Model 1) includes the relative, total, bilateral trade volume as a key variable in addition to the control variables.

Then, we proceed along directional disaggregation, using the upstream and downstream trade connection indicators in Eqs 6 and 7. These indicators are used to build Model 2 in two different ways in order to test the robustness of the results. In Model 2A both directions are included simultaneously, while in Model 2B either the upstream or the downstream variables are entered separately (the suffices -u or -d refer to upstream and downstream specification respectively).

In the next step, we proceed along sectoral disaggregation. As a straight extension of the baseline Model 1, we first include sector-level trade in a bidirectional manner (using $T_{c,t}$ calculated for separate sectors). To check robustness, we use again two different specifications here as well. In Model 3A, we include all sectors simultaneously into the estimations, while Model

3B covers a series of 10 models (for the 10 sectors) where sector-level trade indicators are included separately into every model.

Finally, the most detailed set of models consider sectorally and directionally disaggregated trade indicators. These estimations also come in different setups. Model 4 covers several setups, where only one direction (upstream or downstream) is included. In Model 4B it is only the key variable (plus the controls) which enter the regression, while in Model 4A we include all sectors simultaneously, but in the same direction. Then, Model 5 include both directions simultaneously, with Model 5B having only the key variable on the right hand side, and Model 5A containing all sectors in both directions. Models 4A and 5A again cover 10 different models for the different sectors, while Models 4A and 4B are separately estimated for the two directions. We include control variables in all setups.

Note, that model specifications #A are always more general than specifications #B. However, these labels refer to different generalization under the different cases (adding trade in the opposite direction or adding other sectors in the same direction). The reader may also refer to Table 5 for exact specifications in Models 1 and 2, and to S1 Appendix for the rest of the estimated models.

## Results

### Some descriptive results

Before we move on to the details of the estimated regression models, three sets of descriptive results are presented which complement and introduce the analysis carried out later. First, we present some basic topological properties of the contagion and the relative trade openness networks in order to highlight their fundamental similarities and differences.

For this descriptive analysis, the edges of both networks were compressed in time by adding the elements of the adjacency matrices over the observed time periods:

$$\bar{a}_c = \sum_t (a_{c,t}).$$  (8)

where $a_{c,t}$ is the estimated shock transmission from Eq 3 between country-pair $c$ in time period $t$. Fig 1 shows the shock contagion network, which is formed by the calculated $a_{ij}^{sum}$ elements of the adjacency matrix. The edges of the resulting weighted network indicate the frequency of observing shock transmission from country $i$ to country $j$ over the sample period.

Fig 2 maps the relative trade openness network, which is in the background of the key independent variable in the regression analysis. To build this network we calculated the average of relative trade openness between a country pair $c$ with respect to direction and time:

$$\bar{x}_c = \frac{\sum_t T_{c,t}}{42},$$  (9)

While it is relatively easy to visually explore some differences between the two networks in Figs 1 and 2, a rigorous topological comparison between any two networks is always challenging (see e.g. [60]), but our case is particularly difficult. The links in the shock contagion network express the probability of an event occurring, while the relative trade openness network is a fully interconnected system in which the strength of the edges are important. It follows that even a first glance at Figs 1 and 2 shows striking differences between the two networks. Table 4 extends these visual impressions by listing some topological indicators for the two networks.

While the relative trade openness network is a fully interconnected, complete network, the density of the shock contagion network indicates that the probability of shock propagation

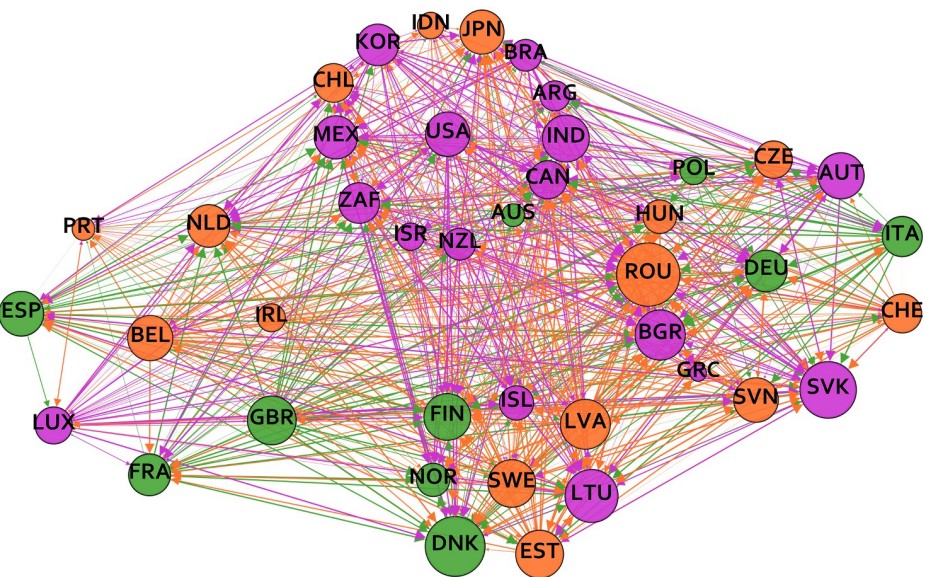

**Fig 1. Weighted shock contagion network.** The nodes of the graph are countries and the directed weighted edges refer to the frequency of observed shock transmission between countries. The color of the nodes reflects the module membership and the size of the nodes refer to the total degrees of countries.

between countries is overall 63.12%. So the first issue is to compare a complete network to a network where not all connections are present. Moreover, both time-compressed networks are weighted, so density does not give a complete picture of the degree of connectivity. Looking at weighted connections and their distribution may give then more insight. We may recall though, that the weight of the links between countries has a completely different meaning in the two networks. As shown earlier, in the case of the shock contagion network, the weights of

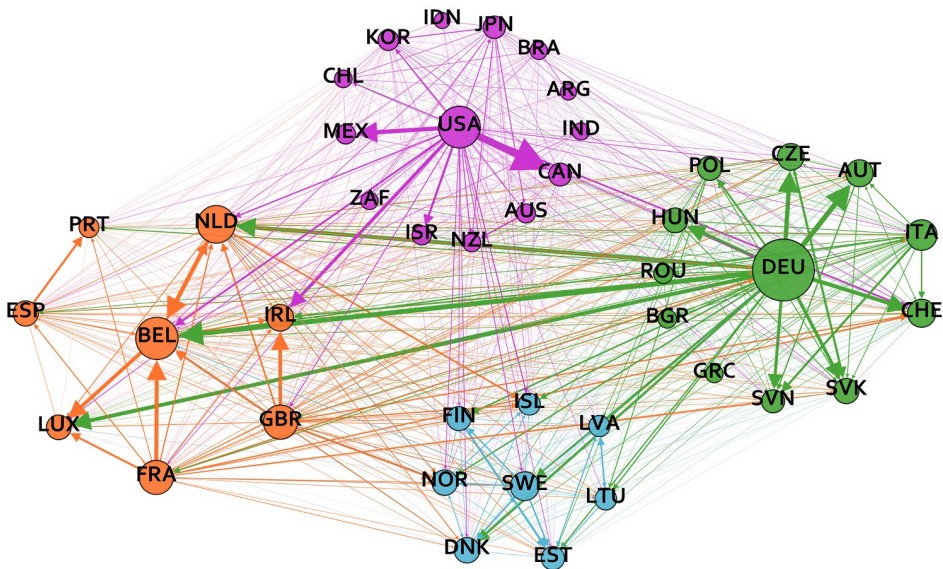

**Fig 2. Relative trade openness network.** The nodes of the graph are countries and the directed weighted edges refer to the average trade volume from country *i* to country *j* relative to the GDP of country *j*. The color of the nodes shows the module membership and the size of the nodes refer to the total degrees of countries.

**Table 4. Global network indicators of the shock contagion and the relative trade openness network.**

|  | Shock contagion | Relative trade openness |
|---|---|---|
| Number of nodes | 42 | 42 |
| Number of edges | 1087 | 1722 |
| Binary density | 0.6312 | 1 |
| Weighted average degree | 1054.619 | 0.4260882 |
| Minimum edge weight | 0 | 0.0000012 |
| Maximum edge weight | 42 | 0.189259 |
| Skewness of edge weights | 0.2116 | 6.2898 |
| Skewness of total weighted degree | -0.0976 | 2.1380 |
| Skewness of weighted outdegree | 0.3204 | 2.9148 |
| Skewness of weighted indegree | 0.6568 | 0.9870 |
| Number of moduls | 3 | 4 |
| Modularity | 0.07184518 | 0.24098753 |

the edges refer to the frequency of shock transmission between a given country-pair. Therefore, the theoretical minimum (no observed shock transmission between the countries) and maximum (shock transmission is observed in all 42 periods) of the weight in this network can be clearly defined. In contrast, weights in the relative trade openness network is not limited from above as trade volumes and outputs can be arbitrary. The minimum observed edge weight in this network is close to zero and the maximum is 0.189 which means that the average of export to and import of country $j$ from country $i$ 18.9% of the GDP of country $j$.

These minimum and maximum edge weights frame then the values of weighted average degree. In the case of the shock contagion network a country can have a maximum weighted degree of 1722 (41 potential neighbors through 42 time periods), while the observed average is cca. 1055. This reflects roughly the same density as for the binary network showing a relatively balanced structure in the background eith respect to high and low degrees. In the relative trade openness network weighted average degree is 0.43 showing that on average a country's trade links with others amount to 43% of their GDP. Taking into account that with the maximum edge weight everywhere this would be 7.76, we can conclude that this network is much less balanced, with the mass of low edge weights outnumbering the high ones.

The distribution of edge weights are reflected by a set of indicators in Table 4. The skewness of the weight distribution is close to a typical normal distribution, for the shock contagion network, which reinforces the previous impression about a relatively balanced structure. In contrast, the relative trade openness network shows a strong positive asymmetry. This means that a few connections have a remarkably high weight, while the majority of the links have typically a low weight—again, in line with the observation from average weighted degrees.

Further substantial differences can be observed in the skewness of the degree distributions. We examine skewness in order to test for the asymmetry in the degree distribution, because standard methods to explore scale-free properties (fitting a power-law) can not be conveniently applied in the case of this small network (42 nodes). The outdegrees and indegrees of the shock propagation network shows slight positive (right) skew, which reflects some asymmetry in the sense that there are less nodes with higher-than-average degrees and more nodes with less-than-average degrees—a property which resembles scale-free structures. In contrast, the skewness of the total degrees is negative, but close to a normal distribution. This means that the role of countries in shock propagation is relatively similar if the direction of links is not taken into account—again, reinforcing the picture which emerged from weighted average

degree. On the contrary, the relative trade openness network shows a different picture. The skewness of degree distributions in this network is strongly positive (strong right skew) which reflects a highly asymmetric network with a few nodes having intensive connections and the majority having relatively weak ones. The skewness of the in-degree distribution is less strong, but still larger than that of the shock contagion network. These results point to the conclusion that the trade openness network is more asymmetric with respect to its structure while the shock contagion network is more balanced.

Finally, we investigated the modularity of the networks in order to capture the extent to which they fall apart into sparsely connected communities with relatively higher within-community density. The cluster-louvain method was used in the igraph package by [61] in R to detect communities in the transformed, undirected networks. The results show that the modularity of the relative trade openness network is higher compared to the shock contagion network. As shown in Fig 2 4 modules are detected in the relative trade openness network. The 4 modules reflect the geographical location of the countries: non-European, Northern European, Central and Eastern European and Western European countries have a higher relative trade openness density compared to the overall network density. Fig 1 shows that 3 communities can be detected in the shock-propagation network, which follow the geographical location less closely. Similar to the network of relative trade openness, the majority of non-European countries are still part of the same community. The exceptions are Chile, Indonesia, Japan and Australia, these countries have more dense shock-contagion links with European countries. However, there are also examples of European countries that are in the module of non-European countries according to their shock contagion links, such as Austria, Luxembourg and Iceland.

These observations about the topological properties of the two networks (shock contagion and relative trade openness) reflect substantial differences between their macroscopic characteristics. However, our goal is to go down to the level of individual connections (directed country-pairs) to evaluate if there is some association between the weight of trade relations and the probability of shock transmission. Panel regression analysis is promising for at least two reasons. First, it exploits the longitudinal nature of the data compared the static and aggregated picture given here and second, it facilitates the comparison of the networks by allowing for an edge-level analysis.

Before going on to the panel regression estimations, as a second descriptive analysis we draw attention to the overall relationship between the degree centrality of the relative trade openness network and the shock contagion network.

Fig 3 shows the log of weighted degrees in the relative trade openness network on the horizontal axis and the log of weighted degree in the shock contagion network on the vertical axis, with every dot representing a country. The positively sloped regression line reflects a slight, but existing positive correlation between the (weighted) degree of the countries in the contagion network and in the relative trade openness network. This means that those countries are more likely to be hit by or be a source of shocks which also maintain more intensive relationships with other countries in trade. This first impression about the connection between calls for further analysis, which we will pursue in the following sections with the help of the previously introduced conditional logit regressions.

After establishing this rough overview of trade and shock contagion, as a third layer of descriptive analysis we take a look at the sectoral decomposition of trade as this is one of the main line of disaggregation in the paper. Fig 4 shows the share of the 10 sectors (as listed in Table 1) in total trade among the sample countries. Every dot represents the share of the given sector in one year, while thick horizontal lines mark the median value. The boxes show upper and lower quartiles around the median and the lines reach out to minimum and maximum

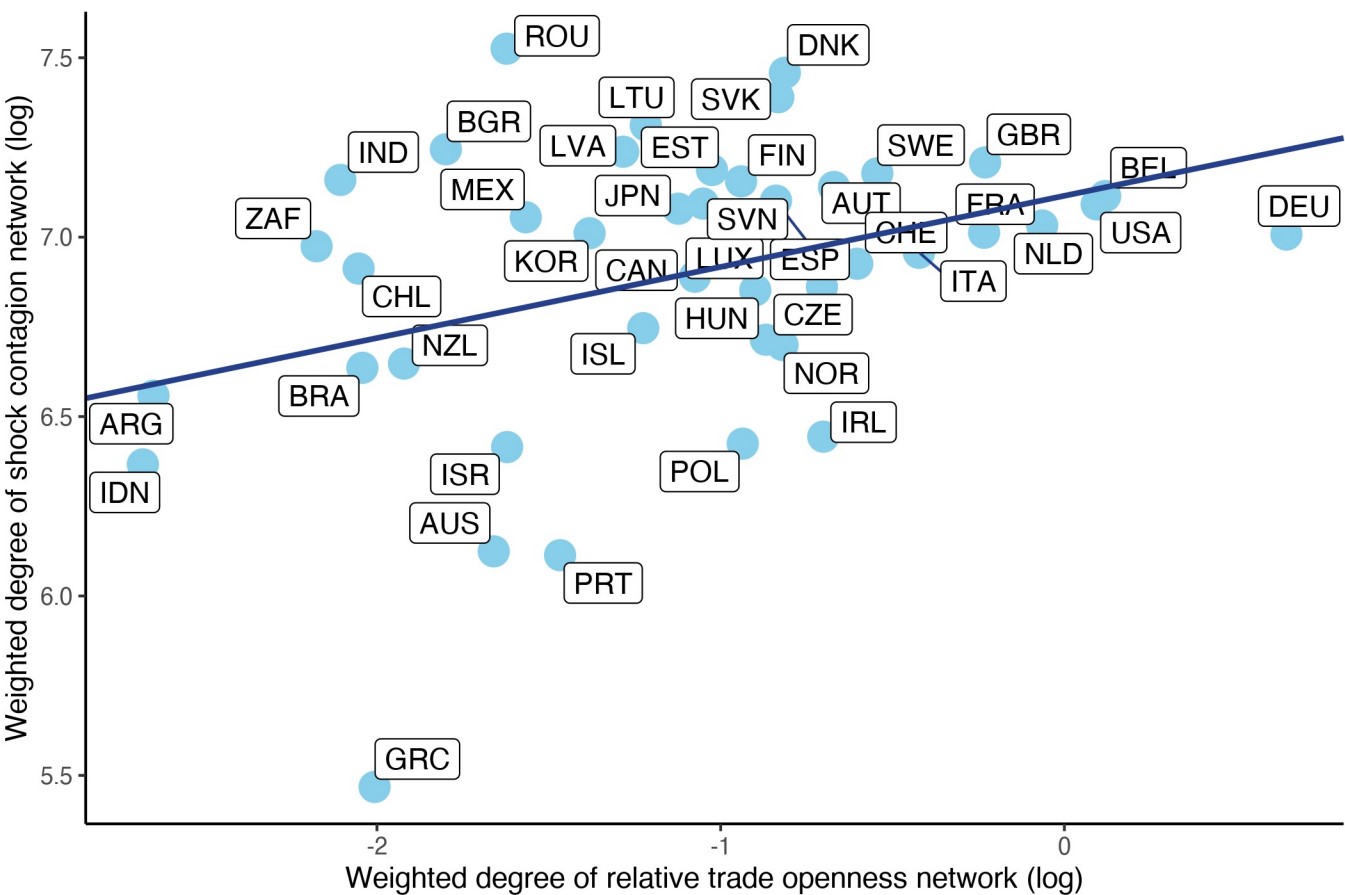

**Fig 3. Association between connectivity in the shock contagion and relative trade openness networks.** The scatter plot shows the correlation between the log of weighted degrees of countries in the relative trade openness network and the log of weighted degrees of countries in the shock contagion network.

values. The picture shows that the machinery and transport equipment sector (7) accounts for 40% of total trade, while the smallest amount is found for beverages and tobacco (1), and animal and vegetable oils, fats, and waxes (4) which give ca. 1% and 0.5% respectively. The picture suggests that these broad sectors play very diverse roles in the trade system. Also, the variability between sectors is much more pronounced than that of the annual trade shares within sectors. Due to the limited availability on inter-country trade in sector number 4, and also supported by its minor role, we exclude this sector from further analysis.

## Models with aggregate trade

Table 5 summarizes the results of the estimations with aggregate trade as the key dependent variable (Models 1, 2 and 3). The table presents the estimated odds-ratios as usual in logit models, together with robust standard errors in the parentheses for the specific variables. While the odds of a given event is the probability of this event to happen relative to the probability of not happening ($p/(1 - p)$ where $p$ is the probability of the event), the odds-ratio measures how a one unit increase in a given independent variable changes the odds of the observed event. An odds ratio of 1.01 in our case thus reflect that a unit change in the given independent variable increases the odds of shock-transmission between two countries by 1%. How this 1% change in the odds then relates to the actual probability of shock-transmission

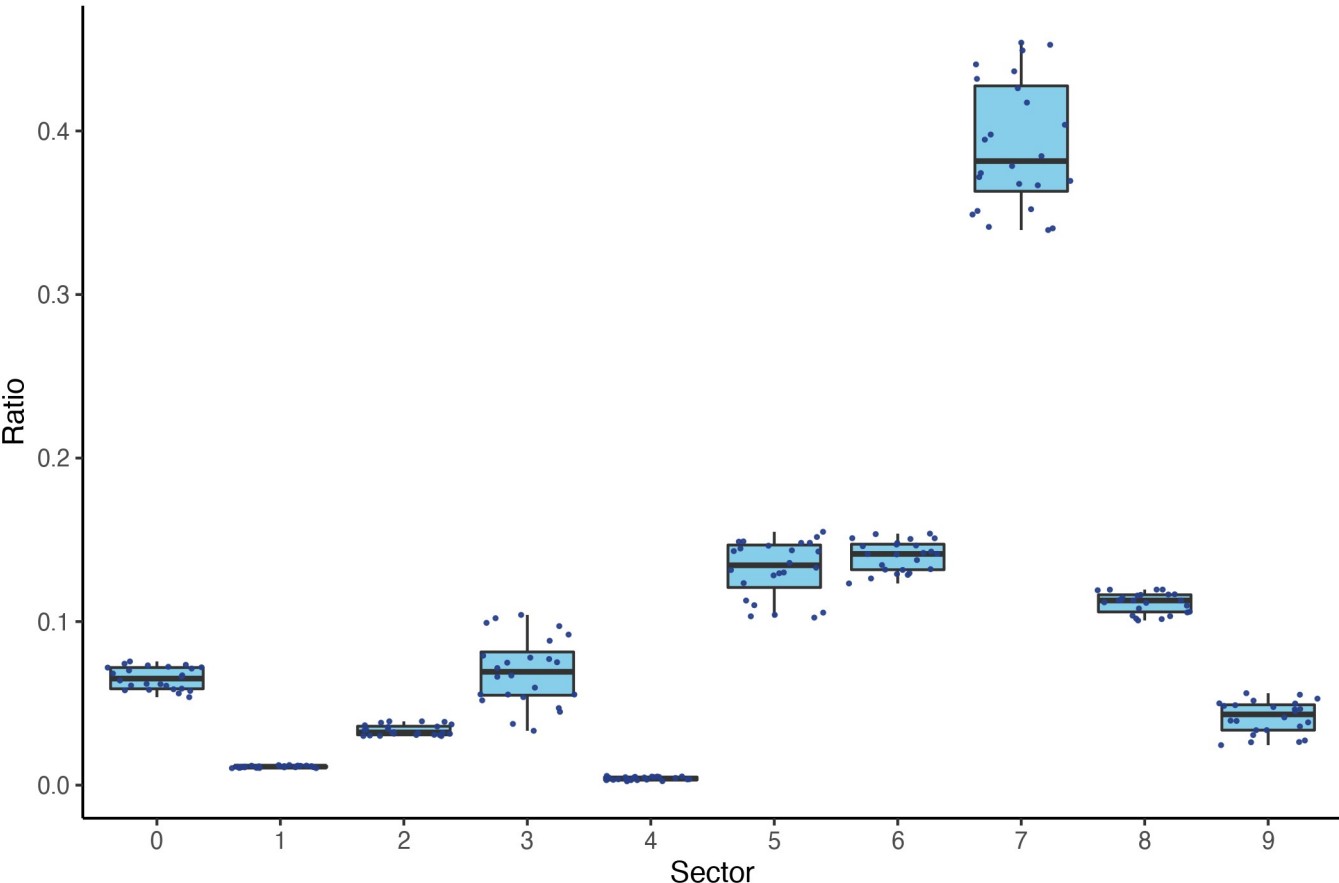

**Fig 4. Share of sectors in total trade.** The box plot shows the share of the 10 broad sectors in total trade among sample countries. Every dot represents the trade share of a given sector in a given year. The horizontal lines show the median of the given sector, the boxes represent the upper and lower quartiles around the median and the vertical lines reach out to minimum and maximum values.

depends on the probability level at which the change happens. If the probability of the event is 1%, then the odds is also very close to 1% (1 to 100), and a 1% increase in the odds corresponds to a cca. 1% increase in the probability of the event, i.e. 0.01 percentage points. If the probability was 50%, then the odds is 1 to 1 (100%), and in this case a 1% increase in the odds corresponds to a 0.5% increase in the probability, which means 0.2 percentage points (a 20-fold difference compared to the previous case). The average probability of shock-transmission is 38.77% in our sample, which means that a 1% odds ratio corresponds to a 0.24 percentage point change in the probability of shock transmission.

In addition to the key independent variables (two-way, upstream or downstream trade openness) we also included the previously described four control variables in the estimations presented in Table 5. These control variables proved to be significant in all aggregate models. The unit of observation in all estimations is a directed country pair $c$ with country $i$ being the origin of the shock transmission while country $j$ being the destination or target. The signs and magnitudes of the estimated coefficients/odds-ratios are robust across the different models. With respect to population (as representing the size of the countries), we see that there is a negative effect for the origin country and a positive one for the destination. This means that in contrary to the gravity principle which would explain a higher probability of shock contagion when either of the two countries is larger, our results suggest that larger countries are more

**Table 5. Results of conditional logit regression.**

| $y_{c,t}$ | Model 1 | Model 2A | Model 2B-u | Model 2B-d |
|---|---|---|---|---|
| $T_{c,t}$ | 1.010309** (0.0052051) | - | - | - |
| $U_{c,t}$ | - | 0.9990693 (0.0071911) | 1.006458 (0.0051588) | - |
| $D_{c,t}$ | - | 1.010078 (0.0086875) | - | 1.009577 (0.0062053) |
| $GDPCap_i$ | 0.8874647*** (0.0226883) | 0.8876793*** (0.0227075) | 0.8907022*** (0.0227005) | 0.8874509*** (0.0225769) |
| $GDPCap_j$ | 1.152534*** (0.0325587) | 1.15319*** (0.0327404) | 1.147854*** (0.0321268) | 1.153347*** (0.0325802) |
| $Pop_i$ | 0.9848761*** (0.0056498) | 0.9852538** (0.0056001) | 0.9851711** (0.0056486) | 0.9851722** (0.0055745) |
| $Pop_j$ | 1.06074*** (0.0107065) | 1.060491*** (0.0106602) | 1.060582*** (0.010671) | 1.060542*** (0.0106651) |
| **Observations** | 38388 | 38388 | 38388 | 38388 |
| **Pseudo $R^2$** | 0.0308 | 0.0312 | 0.0298 | 0.0312 |

Robust standard errors in parentheses,

*** $< 0.001$,

** $< 0.05$,

* $< 0.1$

likely to become the receivers of shocks while smaller countries are more likely to be the origins. From the presented odds-ratios we can conclude that one million inhabitant increases the odds-ratio of shock transmission on the receiver side by roughly 6%, while decreases the odds-ratio on the origin side by 1.5%. In other terms, an additional million of inhabitants increases the probability that a country becomes the receiver of a shock by 1.3–1.5 percentage points on average, and decreases the probability of becoming a shock transmitter by 0.3–0.4 percentage points.

Looking at the coefficients of GDP per capita, we see similar tendencies as with population: the GDP per capita of the origin country negatively affects shock transmission while that of the destination country has a positive effect. This means that shocks are more likely to spread from less developed countries towards more developed ones. In terms of odds ratios, we obtained -11.2% for the transmitting country and 15.3% for the receiving country. This means that a thousand dollar increase in GDP per capita (measured in PPP terms) decreases the probability of shock-transmission to other countries by 2.8 percentage points on average and increases the probability of receiving a shock from others by 3.4 percentage points. Taking into account the unit of measurement here (thousands dollars), this is an economically important effect, meaning that improvements in the level of development (as measured by GDP per capita) considerably changes the position of countries in the shock-contagion network.

These findings with respect to population and per capita GDP are robust across the four model variants presented in Table 5 both in their magnitude and the significance of these effects. Thus our findings reinforce that shocks tend to spread from less developed and/or smaller countries towards more developed and/or larger ones.

In addition to the controls, these estimations primarily focus on the role of trade in shaping shock-transmission between countries. In Model 1 we use two-way aggregate trade, relative to the GDP of country j (the GDP of the shock-receiver country), as defined in Eq 5. We estimate a positive significant effect of this variable at the 5% level, which is represented by an odds-

ratio of 1.01. This means that if exports and/or imports between country $i$ and $j$ increases relative to the GDP of country $j$, the odds of shock transmission from $i$ to $j$ increases by 1%. We measure the trade indicators in units of 0.01 percentage points, thus the 1% odds-ratio means that a slight increase in trade openness by 0.01 percentage point increases the odds of shock transmission between the two countries by 1%. On average, this means a 0.24 percentage point higher probability of shock transmission which can be evaluated as an economically important effect. This result from Model 1 thus establishes a strong relationship between trade the spread of business cycle fluctuations across countries.

While Model 1 provides a general overview of the main relationship under question, versions of Model 2 presented in Table 5 decompose trade links into upstream and downstream connections, according to the trade indicators introduced in Eqs 6 and 7. While the two versions of Model 2B introduces the two directions separately into the estimations, Model 2A simultaneously estimate the effect of upstream and downstream trade connections on shock transmission. Interestingly, these results prove to be insignificant, irrespective of how they enter the estimation. This means that although the two-way trade openness indicator seems to be a significant channel for shock propagation, we cannot conclude that shock propagation occurs along either upstream or downstream mechanisms. This indicates that on the aggregate level, upstream and downstream linkages have to be both present in order to significantly affect shock transmission between countries. Furthermore, a possible reason for this phenomenon is the complexity and diversity of trade relations, which includes many different products and types of channels through which shocks can propagate, possibly neutralizing each other's effects. For example, it may happen that the upstream mechanisms are more dominant in some sectors, while in others the downstream mechanisms play a more important role, producing a blurred effect at the the level of aggregate trade. The results of sectoral trade connections in the next section will allow for a more in-depth analysis of this issue.

## Models with sector-level trade

In this section we turn to the analysis of those models which decompose trade into broad sectors. These sectors were listed in Table 1 and the model variants with sector level trade as the key variable were summarized in the right-hand-side of Table 3. First, we examine sector-level trade without distinguishing between upstream and downstream linkages (Model 3A and Model 3B). Then, we provide estimations with directed trade links (upstream and downstream) separately (Models 4 and 5). In all cases, we experiment with including the sectors separately and together in the estimations, and also with entering upstream and downstream trade separately and together. This results in estimating a considerable amount of different models (entering the sectors separately requires 10 specific estimations for one model variant), so we do not present detailed regression tables here, but focus on the estimated odds-ratios and summarize them in a few diagrams. We include the control variables in all of these models. They remain significant and keep the magnitude of the odds-ratios very robustly across all model variants. The detailed regression results can be found in S1 Appendix.

In Fig 5 we summarize the results of Model 3A and Model 3B. In Model 3A, we include the two-way average trade openness ($T_{c,t}$ in Eq 5) of all sectors, while we have 10 separate estimations for Model 3B with the sectors included one-by-one. As a result, we obtain two estimated coefficients and odds-ratios for all sectors: one in Model A, when all sectors are included and one in model B with the separated estimations. These odds-ratios are summarized in Fig 5 in a way that the shape of the marker reflects the model type (A or B) and the color of the shapes shows the significance (dark means significant at the 10% level, light means non-significant). The position of the markers on the vertical axis shows the estimated odds-ratio for the given

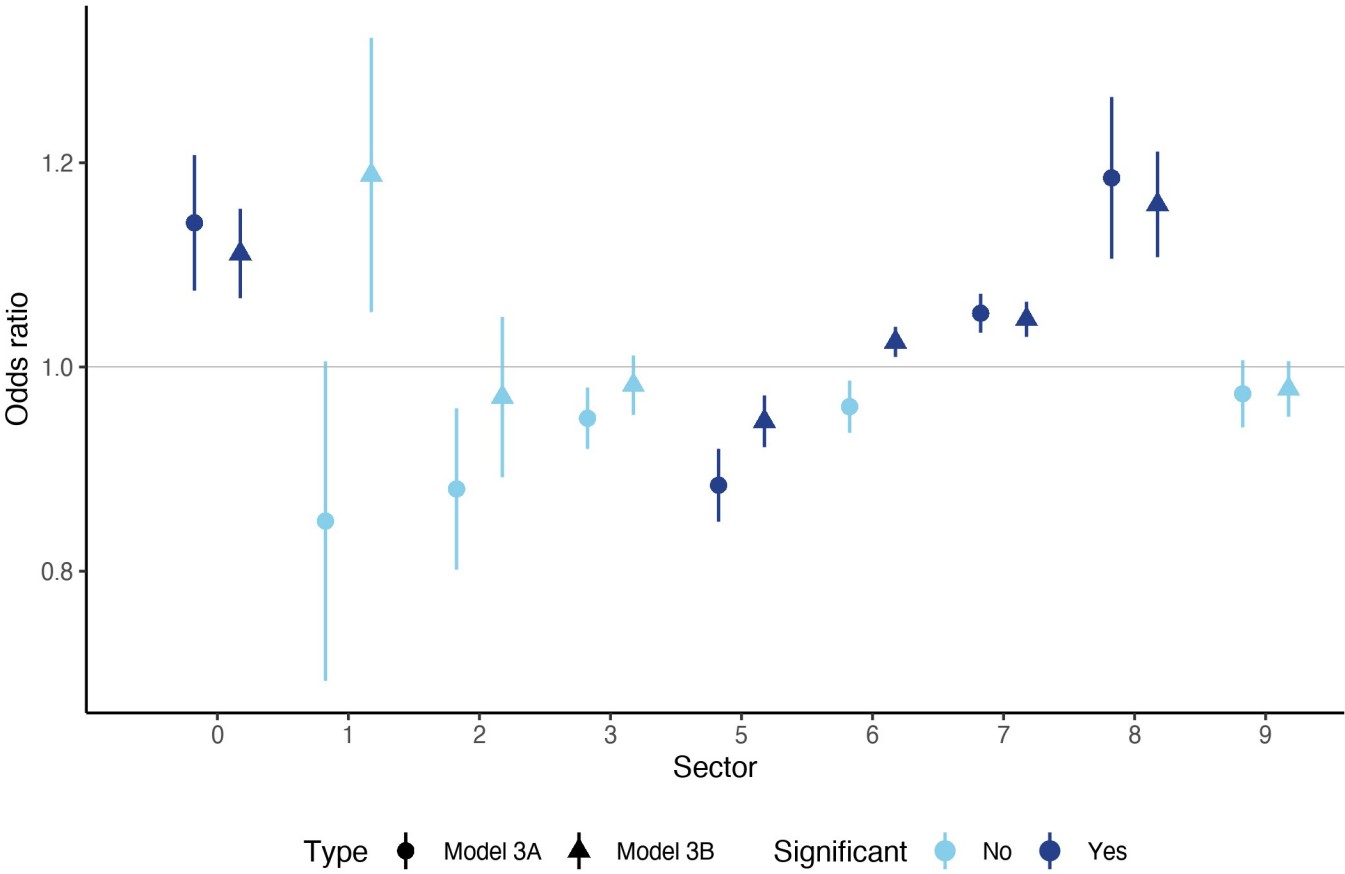

**Fig 5. Results of Model 3A and Model 3B.** Every marker/line represents the estimated odds-ratio of a given sector in a given model type, measured on the vertical axis. The shape of the markers shows the model type. Circles represent estimated odds-ratios in Model 3A, where every sector is included simultaneously, plus the control variables. Triangle markers indicate the estimated odds-ratios in Model 3B, where sectors enter one-by-one. Dark blue markers/lines show significant odds-ratios at the 10% level, light blue markers/colors represent non-significant results. The length of the vertical lines refer to the robust standard errors around the point-estimates. Sectors are grouped along the horizontal axis, marked by their codes (see Table 1 for reference).

specification (the neutral unit odds-ratio is marked by the horizontal grey line) and the length of the vertical lines measures the (robust) standard errors around the point-estimates. Finally, the horizontal axis groups the results according to the sectors (see the sector codes in Table 1.

There are a few general impressions which are visible from this picture. First, the two model variants seem to provide consistent estimations both in terms of the magnitude of the odds-ratios and their significance. The only exception is the sector of manufactured goods classified chiefly by material (6), for which the odds-ratio is only significant in Model 3B, where sectors enter separately. The rest of the sectors where trade connections prove to affect shock contagion are food and live animals (0), chemicals and related products (5), machinery and transport equipment (7) and miscellaneous manufactured articles (8). Second, significant odds-ratios are typically above unity, which shows that these trade channels positively affect shock-contagion: more trade between country $i$ and $j$ relative to the GDP of country $j$ in general, leads to a higher probability of shock contagion from country $i$ to $j$. The only exception in this respect is the sector of chemicals and related products (5) with a significant odds-ratio below one. This result is puzzling, as it suggests that more trade between countries $i$ and $j$ relative to the output of country $j$ makes it less likely that country $i$ affects (causes) the business cycle of country $j$. This seems to be a stabilizing effect through trade the examination of which

requires further work. This sector is very complex, perhaps more diversified than others, as it ranges from chemicals through radioactive materials, medicals and vitamins to cosmetic products like soap. This diversity may act towards less synchronization in macroeconomic outcomes.

If we look at the odds-ratios, we see that the largest value is found for the sector of miscellaneous manufactured articles (8). The estimated odds-ratio for this sector is 18.5% in Model 3A and 15.9% in Model 3B, corresponding to an average of 17.2%. This means that a 0.01 percentage point increase in the openness indicator $T_{c,t}$ in this sector results in a cca. 17% increase in the odds of shock transmission. Taking into account the average shock-transmission probability in our sample, this corresponds to a 3.8 percentage point increase in the latter, which is a considerable effect. The next largest effect is measured for the sector of food and live animals (0) with an average odds-ratio of 12.6%, then machinery and transport equipment (7) follows with 5%. All these are economically important effects. The estimated negative effect in the case of chemicals and related products (5) is also meaningful, with an odds-ratio of -8.5%.

These results coincide with the impressions about the relative shares of sector-level trade in Fig 4. The positive significant effects are found for sectors (0), (6), (7) and (8), which are also the relatively most important ones in the bilateral trade network. Sector (5), which has a negative effect on shock-contagion, is also among the most important ones.

The most detailed picture on shock contagion and trade can be achieved if we further divide trade channels according to their direction. These are done in Models 4 and 5 which are estimated in the same fashion as Model 3: sector level openness indicators may enter separately (Models 4A and 5A) or together (Models 4B and 5B) in the regressions. In this case, however, we have different versions of this model according to the direction included (represented by Models 4), and we also estimate models in which both directions enter simultaneously (Models 5). These model variants are summarized in Table 3.

Fig 6 summarizes the estimated odds-ratios for upstream channels. In these estimations the key variable on the right-hand-side is $U_{c,t}$ in Eq 6, the odds-ratio of which is presented in the figure in a similar manner as in Fig 5. The shade of the markers represents significance, the vertical lines show robust standard errors, while the shape of the markers refers to different model versions. Now we have four model variants for every sector. In Model 4A (circle marker) we include all sectors simultaneously with their upstream connections. In Model 4B (triangle markers) we include only the given sector with its upstream connections. Models 5A and 5B reproduce Models 4A and 4B respectively, but here we also included sector-level downstream connections on the right-hand-side as a robustness check (in Model 5A all downstream sectoral connections are included, while in Model 5B only the downstream connections of the focus sector is included).

The results from Fig 6 reinforce the impression from the results of two-way trade openness (Fig 5) with respect to the sectors which seem important shock-transmission channels. We find significant results for food and live animals (0), chemicals and related products (5), machinery and transport equipment (7) and miscellaneous manufactured articles (8). Also, a reinforcing result is that chemicals and related products (5) prove to be negatively affecting shock-transmission. Finally, mineral fuels, lubricants and related materials are estimated to significantly but negatively affecting shock transmission through upstream channels in contrast to the two-way estimations, while manufactured goods classified chiefly by material (6) looses its (partial) significance observed previously.

With respect to the robustness of the results, we see that it is only the sector of food and live animals (0) which shows positive significant effects across all four model variants. In the case of the other sectors, Model 4A provides consistently significant effects—when all sectors are simultaneously included in the model. When sectors enter individually (Models 4B and 5B) or

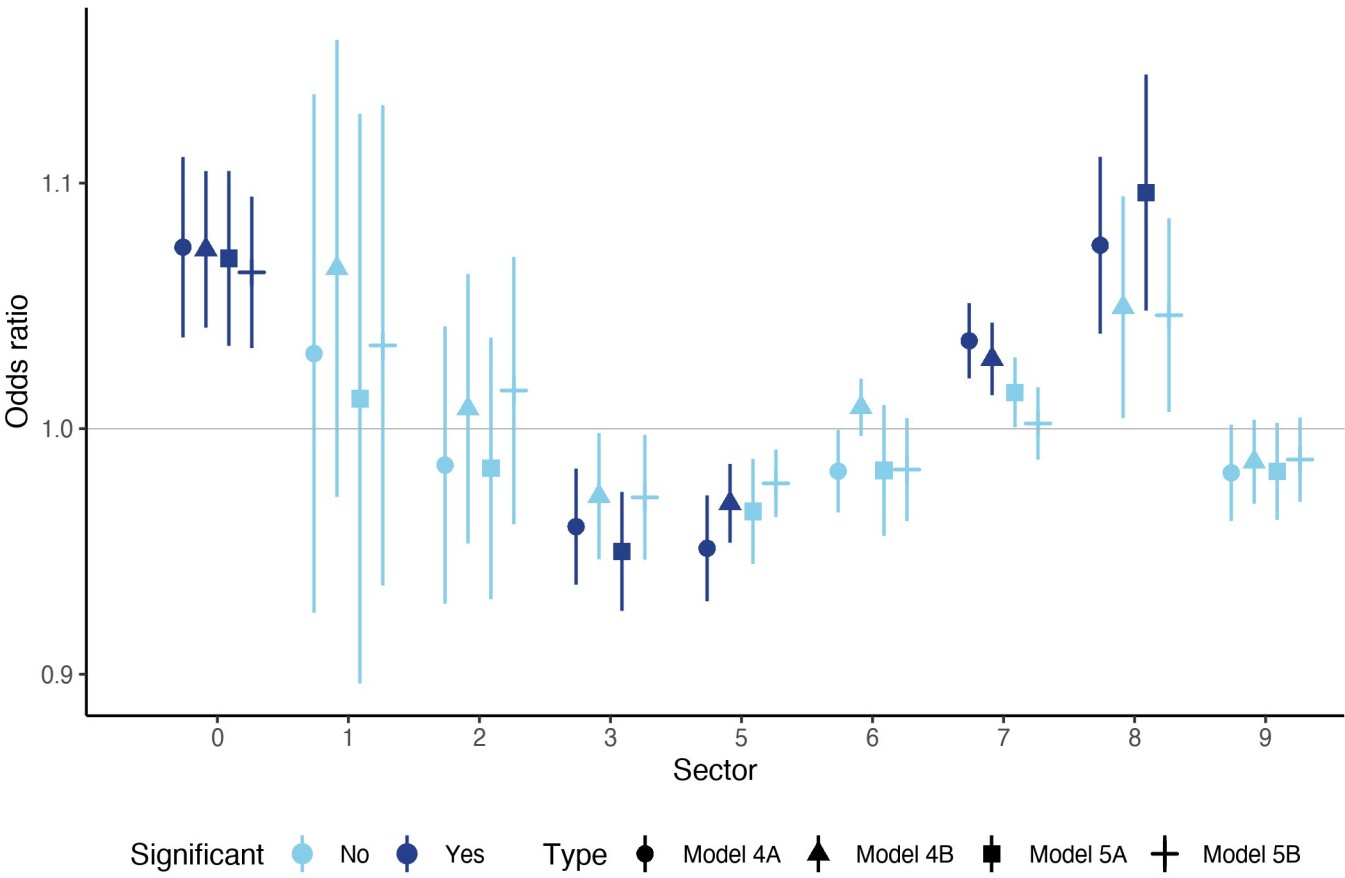

**Fig 6. Results of Model 4A, 4B, 5A and 5B for the upstream variables.** Every marker/line represents the estimated odds-ratio of a given sector in a given model type, measured on the vertical axis. The shape of the markers shows the model type. Circles represent estimated odds-ratios in Model 4A, where every sector is included simultaneously, plus the control variables. Triangle markers indicate the estimated odds-ratios in Model 4B, where sectors enter one-by-one. Squares show the estimated odds-ratios in Model 5A, where sectors enter simultaneously together with downstream connections (all sectors). Horizontal line markers show the estimated odds-ratios in Model 5B, where sectors enter individually, together with their downstream counterparts. Dark blue markers/lines show significant odds-ratios at the 10% level, light blue markers/colors represent non-significant results. The length of the vertical lines refer to the robust standard errors around the point-estimates. Sectors are grouped along the horizontal axis, marked by their codes (see Table 1 for reference).

downstream connections are also included (Models 5A and 5B), significance is lost in some cases.

Looking at the magnitude of the odds-ratios, we again see economically meaningful effects. In the case of the food and live animals (0), the average odds-ratio over the four model types is 7%, which means that the probability of shock transmission is expected to increase by 1.6 percentage points as a result of a 0.01 percentage point increase in the trade indicator $U_{c,t}$. In the case of miscellaneous manufactured articles (8), this amounts to an average odds-ratio of 8.5% for the two significant models while for machinery and transport equipment (7) it is 3.2%. For the two sectors with negative effect the odds ratios are -4% and -4.5% which are also in the economically important range.

Fig 7 contains the results of those estimations which are carried out for downstream sectoral trade indicators as the key variables. The structure of this figure is the same as that of Fig 6, but the results are markedly different—which shows that upstream and downstream trade channels affect shock contagion differently. While in the upstream channel the sector of food and live animals (0) was found to be robustly and significantly affecting shock contagion, in

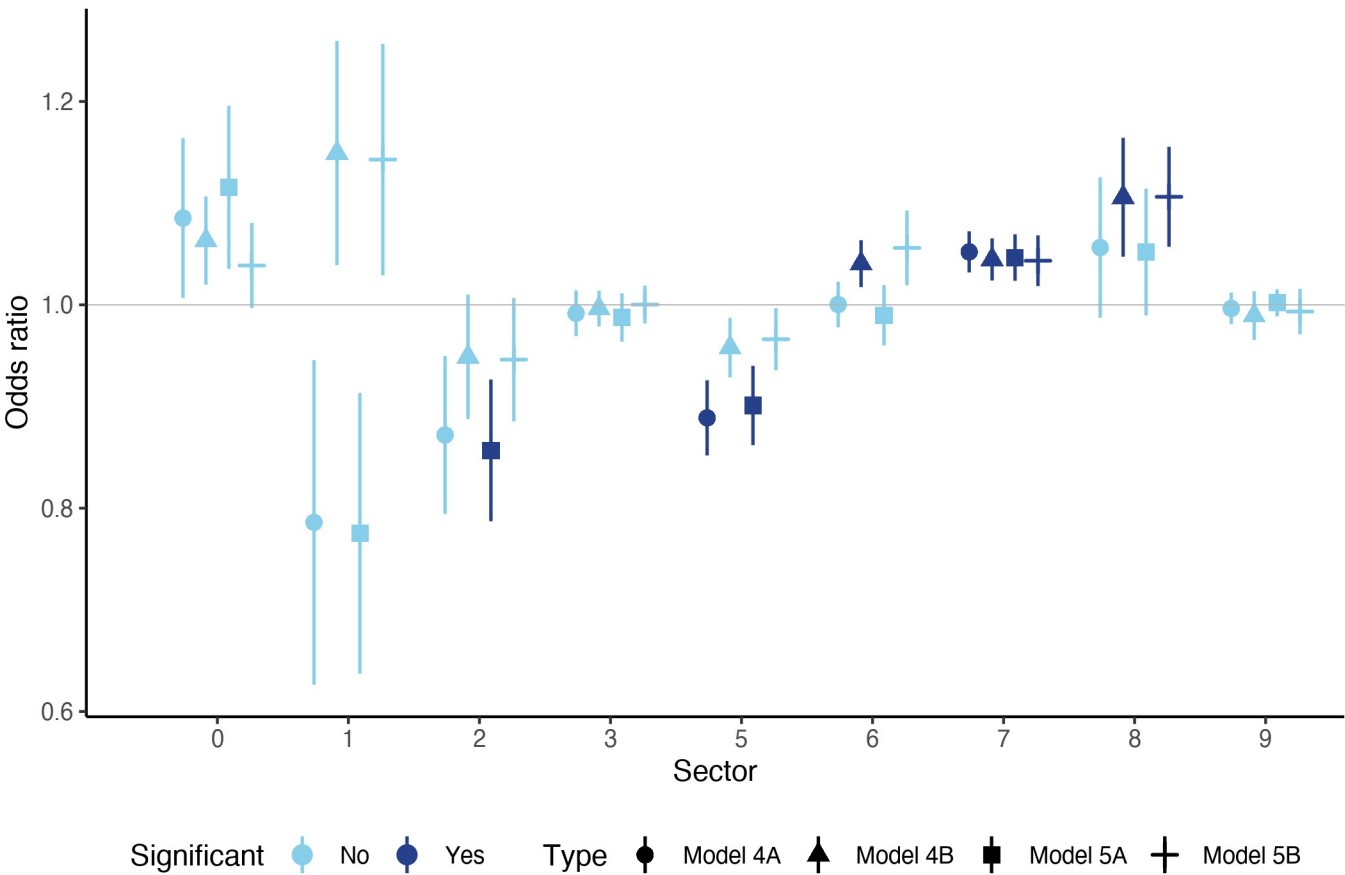

**Fig 7. Results of Model 4A, 4B, 5A and 5B for the downstream variables.** Every marker/line represents the estimated odds-ratio of a given sector in a given model type, measured on the vertical axis. The shape of the markers shows the model type. Circles represent estimated odds-ratios in Model 4A, where every sector is included simultaneously, plus the control variables. Triangle markers indicate the estimated odds-ratios in Model 4B, where sectors enter one-by-one. Squares show the estimated odds-ratios in Model 5A, where sectors enter simultaneously together with upstream connections (all sectors). Horizontal line markers show the estimated odds-ratios in Model 5B, where sectors enter individually, together with their respective upstream counterparts. Dark blue markers/lines show significant odds-ratios at the 10% level, light blue markers/colors represent non-significant results. The length of the vertical lines refer to the robust standard errors around the point-estimates. Sectors are grouped along the horizontal axis, marked by their codes (see Table 1 for reference).

the downstream channel we do not find a significant effect for this sector in either of the model variants. On the other hand, the sector of machinery and transport equipment (7) turns out to have a positive significant effect in the downstream channel, but this result is very robust now across the different model types. We also find a positive significant effect for miscellaneous manufactured articles (8), but not in all model types. In the case of chemicals and related products (5) the negative effect remains also in the downstream channel, but this finding is not very robust, similarly to the previous cases with upstream and two-way channels. We find significant effects in the case of the sectors (2) and (6), but only in one model variant, so we do not count these as robust results.

The magnitude of the odds-ratios for the significant sectors turn to be economically meaningful in these cases as well. For machinery and transport equipment (7) the estimated average odds-ratio is 4.7% which means a cca. 1 percentage point increase in the probability of shock-transmission as a result of a 0.01 percentage point increase in the trade openness indicator $D_{c,t}$. The odds-ratio is even higher, 10.6% for miscellaneous manufactured articles (8)—although these results are not that robust. In the case of chemicals and related products (5), the odds-ratio is also quite large, -10.5%.

## Discussion and conclusion

In this paper we intended to re-examine the relationship between trade openness and business cycle synchronization by providing a relatively detailed account of this issue compared to previous studies. These extensions manifested in three areas. First, we used a Granger-causality approach to identify synchronization which allows a better overview on how macroeconomic shocks transmit from one country to another, compared to cross-correlation studies. Second, we used a sectoral breakdown of trade in order to gain more insights about the specific sectors/industries which are in the background of shock contagion. Third, and in addition to the sectoral approach, we distinguished between upstream and downstream connections, which allowed for a directed approach in the analysis with respect to the trade relations—in line with the directed nature of the contagion network based on Granger causality.

We used conditional logit regressions to estimate the effect of trade openness on the probability of shock-transmission. The unit of observation was directed country pairs for time periods as rolling time windows. Trade openness is measured by (directed) trade volumes relative to the output of the country which counts as the shock-receiver within the country-pair. We gradually added details to the regression by focusing on two-way (undirected) and aggregate trade first and then adding the directional (upstream and downstream) and sectoral dimensions to the analysis. The results from the regression estimations can be summarized as follows.

- There is evidence that trade connections are conducive to shock-transmission, but our results reveal sectoral and directional specificities as we add more detail to the analysis.

- Aggregate two-way trade has a positive significant effect on shock contagion, but significance is lost if we divide aggregate trade into upstream and downstream directions.

- When two-way trade is disaggregated into broad sectors, we find that the overall positive effect can be traced back to some sectors. More specifically, food and live animals (0), machinery and transport equipment (7) and miscellaneous manufactured articles (8) are found to be significant channels for shock-transmission. On the other hand, chemicals and related products (5) seem to have a negative effect, thus contributing more to the diversification of risks rather than spreading shocks.

- When we disaggregate sectoral trade into upstream and downstream directions, these results are further shaded to some extent. Food and live animals (0) prove to be a significant shock-transmission channel only upstream, while machinery and transport equipment (7) and miscellaneous manufactured articles (8) seem to transmit shocks both upstream and downstream. However, the most robust results are obtained for Food and live animals (0) in the upstream and machinery and transport equipment (7) in the downstream channels.

- The negative effect of chemicals and related products (5) on shock-propagation is visible in both upstream and downstream channels, but these results are not very robust.

- Checking the magnitude of the estimated odds-ratios we found values between 1% and 17%. Taking into account the unit of the key dependent variable, which is measured in 0.01 percentage points these values prove to be economically important: a 0.01 percentage point increase in the trade openness (trade volume relative to GDP) increases the probability of shock-contagion by 0.24–3.8 percentage points, depending on the actual value of the estimated odds-ratio.

- In addition to the role of trade channels in shock propagation, our results also show that the size of the countries (population) and their level of development (GDP per capita) shape

shock propagation. Shock-transmission is found to be more likely from smaller and/or less developed countries to larger and/or more developed ones.

As a result, we further refine previous studies examining the link between trade openness and business cycle synchronization. The results of this literature are not unambiguous, some pointing to a moderate positive effect [40–43] while others arguing a less pronounced role for trade [44–47]. The results presented in this study contribute to this literature in two interconnected ways. First, by revealing that the overall positive effect found in aggregate two-way trade hides diverse behavior in specific trading sectors as well as upstream and downstream channels. Second, by pointing out important and less important channels in this background. In the latter respect we see that while some sectors are not significant channels of shock-transmission in either directions, upstream channels seem to be important in agriculture while downstream channels dominate machinery and other manufactures. Also, there are sectors (chemicals and related products) trade in which negatively affects shock-transmission.

Apart from the results, our approach has clear limitations as well, which point to avenues for further research and clarification. First, this study only takes into account trade in goods as channels of shock-transmission, and excludes services. Also, financial linkages may play an equally important role in shock-propagation which is not tackled by this study. In addition to these limits in data coverage, we may also point out that shock-contagion as the dependent variable is estimated on a quarterly basis while annual trade data is used as independent variables. Although these were aligned in a convincing way, more accurate results would be available if quarterly trade data were used—but in this case the time coverage has to be shortened. Using rolling time windows we are able to considerably extend the time dimension of the panel, but it comes at the risk of merging otherwise heterogeneous time-periods into the same time window.

While the use of the fixed-effects panel model in this study reduces the risk of bias arising from omitted variables, it has limitations in handling potential endogeneity in the data. Our estimation procedure builds on the assumption that trade relationships are exogenous to shock transmission, so the latter do not affect the former, however, we can not rule this option out. This reverse causality would mean that shocks hitting the economies affect their economic processes, which in turn affect the trade patterns. While we can not rule this effect out, we argue that the assumption of trade links channelling shocks is stronger and are in line with economic models as already discussed under the upstream and downstream channels of shock propagation in the Results section. At the same time, the potential mechanisms behind the reverse causality are indirect and unfold over much longer periods of time. A further methodological limitation of the paper is that a considerable portion of the observations are dropped from the estimations due to the lack of variation in the dependent variable. Although we argued that the results are not biased due to this problem, but a more accurate picture would emerge if other econometric techniques were used capable of handling this issue. Although the Granger-causality approach is more powerful in identifying actual shock-transmission between business cycles, a clear advantage of cross-correlations in this respect would be to provide a continuous dependent variable, which eliminates the previous problem.

## Supporting information

**S1 Appendix. Regression results.** This document contains the detailed conditional logit regression results (tables) for every model version that was estimated. The regression results are ordered according to Table 3.
(PDF)

**S1 Table. Data table.** This table contains the applied database. The table provide trade volumes, the calculated openness indicators and control variables for all country-pairs and periods in the sample.
(DTA)

**S2 Table. Information.** This table contains additional information on the data in S1 Table together with a short description of the contents.
(CSV)

**S1 Fig. Additional figure.** The figure is a histogram describing the distribution of observed shock-transmission events within country-pairs. For all (directed) country pairs we have 42 observations (time periods). The horizontal axis measures the number of time periods (from 1 to 41) and the bars at every value represent the number of country-pairs for which shock-transmission is observed exactly that many times as the number of time periods on the horizontal axis. We excluded those country-pairs from the figure for which we either do not observe shock transmission at all (at 0) or we observe transmission in every period (at 42), as these country-pairs are also dropped from the conditional logit regressions.
(TIF)

## Author Contributions

**Conceptualization:** Zita Iloskics, Tamás Sebestyén, Erik Braun.

**Data curation:** Zita Iloskics.

**Formal analysis:** Zita Iloskics, Erik Braun.

**Funding acquisition:** Tamás Sebestyén.

**Investigation:** Zita Iloskics, Tamás Sebestyén, Erik Braun.

**Methodology:** Zita Iloskics, Tamás Sebestyén, Erik Braun.

**Project administration:** Zita Iloskics, Tamás Sebestyén, Erik Braun.

**Software:** Zita Iloskics.

**Validation:** Zita Iloskics, Tamás Sebestyén, Erik Braun.

**Visualization:** Zita Iloskics.

**Writing – original draft:** Zita Iloskics, Tamás Sebestyén, Erik Braun.

**Writing – review & editing:** Zita Iloskics, Tamás Sebestyén, Erik Braun.

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
