## [Decision Letter · Decision Letter 0]

5 May 2021

PONE-D-21-08862

Shock propagation channels behind the global economic contagion network. The role of economic sectors and the direction of trade.

PLOS ONE

Dear Dr. Iloskics,

Thank you for submitting your manuscript to PLOS ONE. After careful consideration, we feel that it has merit but does not fully meet PLOS ONE’s publication criteria as it currently stands. Therefore, we invite you to submit a revised version of the manuscript that addresses the points raised during the review process.

We look forward to receiving your revised manuscript.

Kind regards,

Hocine Cherifi

Academic Editor

PLOS ONE

Journal Requirements:

Thank you for stating the following financial disclosure:

X

Thank you for stating the following in your Competing Interests section: 

X

Reviewers' comments:

Reviewer's Responses to Questions

**Comments to the Author**

1. Is the manuscript technically sound, and do the data support the conclusions?

Reviewer #1: Partly

Reviewer #2: Yes

2. Has the statistical analysis been performed appropriately and rigorously? 

Reviewer #1: Yes

Reviewer #2: Yes

3. Have the authors made all data underlying the findings in their manuscript fully available?

Reviewer #1: Yes

Reviewer #2: Yes

4. Is the manuscript presented in an intelligible fashion and written in standard English?

Reviewer #1: Yes

Reviewer #2: Yes

5. Review Comments to the Author

Reviewer #1: The title and the introduction suggests an interesting research to apply network analysis to economics. The data collection and pairwise analysis looks good as a technical merit. However, after detailed reading, there are some technical issues that prohibits reviewer's recommendation for publication in its current form.

- Network science suggests a new technical frontier to study economics, particularly international trade, and there are many existing papers looking into this direction. For contagion networks, in addition to the links as this paper paying most attention, network analysis of directed or bi-directed links could well tell a lot of stories behind.

- The regression methods including logit regression, are rather standard. Therefore, these methods can only tell behavior over links, without a global comprehension as a network. To explore synchronization in a contagion network, the analytical tools presented in this paper are not sufficient.

- The openness in trade is equivalently measured by the incoming and outgoing ratio of a network node, which could not fully comprehend the behavior according to network science (or (Google) search in the network).

Due to reviewer's limited expertise in economics and trade, the contributions to economic behavior could not be well judged, though this reviewer could not identify new finding. In light of limited data analytical tools and no novel application of these tools in this paper, this reviewer suggests authors seek publication of this paper in the economy journals, rather than PLOS ONE.

Reviewer #2: The authors examine the relationship between trade openness and business cycle synchronisation. More specifically they investigate synchronisation evaluating Granger causality for GDP time series, then the authors study trade at a sector level, distinguish between production (downstream) shocks and demand (upstream) shocks and study their relationship with trade openness.

The analysis is solid and the question of relevance and I recommend the paper for publication after a minor revision.

My main comment is related to clarifying the quantities and concepts that are being used by the authors. In particular, the authors introduce the concept of shock contagion, Granger causality, trade openness, synchronisation, upstream and downstream contagion, and substantially they evaluate two networks, one (static) extracted from GDP time series through Granger causality and one multi-layer networks (dynamic) related to trade between different countries in different sectors.

On this point, I think the authors should try to present an exploratory analysis of these two networks, describing their basic network properties. This would also allow to put the regression results in perspective.

Secondly, I think the authors should better clarify how the concepts they define are related to the two networks, e.g. shock contagion is derived from GDP time series but it is referred to also as synchronisation, then they use the term upstream and downstream shocks, referring to trade relationships, but it is not entirely clear how these labels are assigned to individual links. I believe the authors have tried to clarify these terms but I think the overall structure of the methodology could be improved.

Further, if the authors introduced a more comprehensive general model which encompassed Model#A and Model#B the paper would probably be more accessible and intuitive, but this is a personal suggestion rather than a request for a revision. On this point the attempt to clarify the multiple tested models in Table 3 is partially successful.

6. PLOS authors have the option to publish the peer review history of their article (what does this mean?). If published, this will include your full peer review and any attached files.

Reviewer #1: No

Reviewer #2: No

---

## [Author Response · Author response to Decision Letter 0]

5 Jul 2021

First of all, we would like to thank for all the valuable and insightful comments of the two reviewers. We have learned a lot from them, and reviewed the manuscript accordingly, hoping that it gained a lot in clarity and the interpretation of the results. In what follows, we reflect on the reviewers’ comments one-by-one and also point to the respective changes made to the manuscript. The reviewer comments are typed in italic while our answers are in normal font.

Reviewer #1

- 'Network science suggests a new technical frontier to study economics, particularly international trade, and there are many existing papers looking into this direction. For contagion networks, in addition to the links as this paper paying most attention, network analysis of directed or bi-directed links could well tell a lot of stories behind.'

Thanks for this note. In order to emphasize this aspect, we inserted some references about network approaches in the analysis of shock contagion into the introduction (~ line 50).

- 'The regression methods including logit regression, are rather standard. Therefore, these methods can only tell behavior over links, without a global comprehension as a network. To explore synchronization in a contagion network, the analytical tools presented in this paper are not sufficient.'

Indeed, the main methodological vehicle in this study is the logit regression which focuses on edge-level association between phenomena. In order to augment this focus, we inserted a topological analysis of the two networks under consideration: the shock-contagion network and the relative trade openness network. This analysis includes visualization as well as a comparison among the topological properties of these networks as much as possible (the content of the edge-weights establish some limitation to this comparison). We believe that this additional analysis enriches the descriptive part of the paper, makes the study more complete in terms of network analysis and also provides a good background for understanding the right-hand side and the left-hand side key variables in the regression models.

- 'The openness in trade is equivalently measured by the incoming and outgoing ratio of a network node, which could not fully comprehend the behavior according to network science (or (Google) search in the network).'

The chosen trade openness measure (which we consistently relabeled as relative trade openness in the revised manuscript) is built on standard openness measures. Apart from distinguishing between upstream and downstream connections, this construction that we use implies a direction to trade relationships as it measures the relative importance of a given trade connection for the target country, where the latter is defined on the basis of the shock contagion network. This way, we rule out size effects from the estimations and put more weight to those connections which are intensive relative to the size of the shock-receiving country. We emphasized this approach and the rationale behind our choice in the section Model variants: upstream and downstream channels, sectors. 

- 'Due to reviewer's limited expertise in economics and trade, the contributions to economic behavior could not be well judged, though this reviewer could not identify new finding. In light of limited data analytical tools and no novel application of these tools in this paper, this reviewer suggests authors seek publication of this paper in the economy journals, rather than PLOS ONE.'

We emphasized both in the abstract and the conclusion that the main contribution of the paper is to reveal the finer details of shock propagation channels than just looking at aggregate bilateral trade, and point out those sectors and directions which makes it more likely for shocks to spread between countries. Also, we feel that the included topological analysis made the paper more relevant from the complex systems perspective as well.

Reviewer #2

- 'My main comment is related to clarifying the quantities and concepts that are being used by the authors. In particular, the authors introduce the concept of shock contagion, Granger causality, trade openness, synchronisation, upstream and downstream contagion, and substantially they evaluate two networks, one (static) extracted from GDP time series through Granger causality and one multi-layer networks (dynamic) related to trade between different countries in different sectors.

On this point, I think the authors should try to present an exploratory analysis of these two networks, describing their basic network properties. This would also allow to put the regression results in perspective.'

Thanks for this note and suggestion. Indeed, the analysis have exclusively focused on the logit regression and little has been exposed about the two networks that are in the background of the left and right hand sides of the regression equation. In order to fill this gap, we inserted a topological analysis of the two networks into the section of descriptive results. This analysis includes visualization as well as a comparison among the topological properties of these networks as much as possible (the content of the edge-weights establish some limitation to this comparison). We believe that this additional analysis enriches the descriptive part of the paper, makes the study more complete in terms of network analysis and also provides a good background for understanding the right-hand side and the left-hand side key variables in the regression models.

-'Secondly, I think the authors should better clarify how the concepts they define are related to the two networks, e.g. shock contagion is derived from GDP time series but it is referred to also as synchronisation, then they use the term upstream and downstream shocks, referring to trade relationships, but it is not entirely clear how these labels are assigned to individual links. I believe the authors have tried to clarify these terms but I think the overall structure of the methodology could be improved.'

Thanks for this observation. We tried to render the exposition of the methods and indicators more concise by modifying the wording and adding clarifying sentences. Also, we made a consistent labeling of shock contagion network and links for the dependent variable and relative trade openness network and links for the key independent variables. Also, we clarified the Introduction in order to distinguish between synchronization, shock contagion and trade links.

- 'Further, if the authors introduced a more comprehensive general model which encompassed Model#A and Model#B the paper would probably be more accessible and intuitive, but this is a personal suggestion rather than a request for a revision. On this point the attempt to clarify the multiple tested models in Table 3 is partially successful.'

Thanks for this note. We would like to emphasize that models labelled with A are always more general than models labelled with B. In this sense, estimating a model with specifications #A and #B included would mean running model #A. Also, in the case of aggregate and sector level estimations #A and #B models do not refer to the same (or similar) specifications: for example, Model 2B includes only the key variable on the RHS plus the controls, while Model 2A adds opposite direction (upstream to downstream and vice versa). However, Model 3B includes only the bidirectional sector-level trade on the RHS (plus controls), while Model 3A adds all other sectors to the RHS as well, but in the same direction. We added a similar note to the discussion of Table 3 and referred to the regression tables to clarify the given specifications in the background.

---

## [Decision Letter · Decision Letter 1]

2 Aug 2021

PONE-D-21-08862R1

Shock propagation channels behind the global economic contagion network. The role of economic sectors and the direction of trade.

PLOS ONE

Dear Dr. Iloskics,

Thank you for submitting your manuscript to PLOS ONE. After careful consideration, we feel that it has merit but does not fully meet PLOS ONE’s publication criteria as it currently stands. Therefore, we invite you to submit a revised version of the manuscript that addresses the points raised during the review process.

We look forward to receiving your revised manuscript.

Kind regards,

Hocine Cherifi

Academic Editor

PLOS ONE

Journal Requirements:

Reviewers' comments:

Reviewer's Responses to Questions

**Comments to the Author**

1. If the authors have adequately addressed your comments raised in a previous round of review and you feel that this manuscript is now acceptable for publication, you may indicate that here to bypass the “Comments to the Author” section, enter your conflict of interest statement in the “Confidential to Editor” section, and submit your "Accept" recommendation.

Reviewer #1: (No Response)

Reviewer #2: All comments have been addressed

2. Is the manuscript technically sound, and do the data support the conclusions?

Reviewer #1: Partly

Reviewer #2: Yes

3. Has the statistical analysis been performed appropriately and rigorously? 

Reviewer #1: Yes

Reviewer #2: Yes

4. Have the authors made all data underlying the findings in their manuscript fully available?

Reviewer #1: Yes

Reviewer #2: Yes

5. Is the manuscript presented in an intelligible fashion and written in standard English?

Reviewer #1: Yes

Reviewer #2: Yes

6. Review Comments to the Author

Reviewer #1: The authors respond my issues, though some slight concerns remaining. The aspects about causality and effect of regression (logic regression has limitation in high-dimensional data) remains unclear to this reviewer. In light of the spirit of PLOS ONE, this manuscript has some interesting point and can be publishable after improvement on above two aspects.

Reviewer #2: (No Response)

7. PLOS authors have the option to publish the peer review history of their article (what does this mean?). If published, this will include your full peer review and any attached files.

Reviewer #1: No

Reviewer #2: No

---

## [Author Response · Author response to Decision Letter 1]

16 Sep 2021

First, we would like to thank for the further valuable comments of the reviewer. We considered these comments and reviewed the manuscript accordingly, hoping that the concerns of Reviewer#1 have been addressed. We identified three layers in which our paper can address the issues raised by the reviewer and refer to these below, also pointing to the respective changes made to the manuscript. 

Comment of Reviewer #1: 'The authors respond my issues, though some slight concerns remaining. The aspects about causality and effect of regression (logic regression has limitation in high-dimensional data) remains unclear to this reviewer. In light of the spirit of PLOS ONE, this manuscript has some interesting point and can be publishable after improvement on above two aspects.'

We identify a specific, primarily technical layer mentioned in the parentheses of the reviewer comment: this refers to the limitations of using logistic regression with high-dimensional data. With respect to this issue, we would like to note that high-dimensionality does not arise in our estimations, as the number of variables included in any of the regressions is magnitudes lower than the number of observations. We pointed to this in the manuscript, by inserting the following paragraph (~line 210). 

‘Recent studies also call attention on the limitations of applying logistic regression in the case of high-dimensional data [59]. However, this limitation is not binding in this study, since the regression analysis employs 37 variables for approximately 38,000 observations, meaning that the condition for high-dimensionality does not apply. Moreover, these variables are not used simultaneously in the estimations (the most extensive estimation includes 24 variables and runs with 34,152 observations).’

As a second layer of the reviewer comment, we identified possible limitations with respect to potential endogeneity in the regressions, as the assumed effect (trade channels affect shock transmission) may be reversed (shocks affect trade patterns). We refer to this limitation in the discussion part, among other limitations of the study, by inserting the following paragraph (~756).

‘While the use of the fixed-effects panel model in this study reduces the risk of bias arising from omitted variables, it has limitations in handling potential endogeneity in the data. Our estimation procedure builds on the assumption that trade relationships are exogenous to shock transmission, so the latter do not affect the former, however, we can not rule this option out. This reverse causality would mean that shocks hitting the economies affect their economic processes, which in turn affect the trade patterns. While we can not rule this effect out, we argue that the assumption of trade links channelling shocks is stronger and are in line with economic models as already discussed under the upstream and downstream channels of shock propagation in the Results section. At the same time, the potential mechanisms behind the reverse causality are indirect and unfold over much longer periods of time.’

Finally, a third layer of the reviewer comment drove us towards potential controversies around the concept of Granger causality on which shock transmission is based in our paper. Although this is a widely accepted tool in time series analysis, there are clear limitations which we mentioned in the manuscript when introducing the method, by inserting the following paragraph (~line 131).

‘While Granger causality tests are widely used in time series analysis as a method to identify casual relationships between observed variables, it is important to emphasise its limitations. Although Granger causality is suitable for detecting co-movement between time series and reflects the sequence of events, these identified cross-period co-movements does not always represent real causal relationships. The opposing views on the existence of actual causality and the different philosophical approaches to its identification are reviewed by e.g. [60, 61].’

---

## [Editor Report · Decision Letter 2]

24 Sep 2021

Shock propagation channels behind the global economic contagion network. The role of economic sectors and the direction of trade.

PONE-D-21-08862R2

Dear Dr. Iloskics,

We’re pleased to inform you that your manuscript has been judged scientifically suitable for publication and will be formally accepted for publication once it meets all outstanding technical requirements.

Kind regards,

Hocine Cherifi

Academic Editor

PLOS ONE
---

## [Editor Report · Acceptance letter]

9 Oct 2021

PONE-D-21-08862R2 

Shock propagation channels behind the global economic contagion network. The role of economic sectors and the direction of trade. 

Dear Dr. Iloskics:

I'm pleased to inform you that your manuscript has been deemed suitable for publication in PLOS ONE. Congratulations! Your manuscript is now with our production department. 

Kind regards, 

on behalf of

Professor Hocine Cherifi 

Academic Editor

PLOS ONE